# The Multidisciplinary Approach in Stage III Non-Small Cell Lung Cancer over Ten Years: From Radiation Therapy Optimisation to Innovative Systemic Treatments

**DOI:** 10.3390/cancers14225700

**Published:** 2022-11-20

**Authors:** Alessandra Ferro, Matteo Sepulcri, Marco Schiavon, Elena Scagliori, Edoardo Mancin, Francesca Lunardi, Gisella Gennaro, Stefano Frega, Alessandro Dal Maso, Laura Bonanno, Chiara Paronetto, Francesca Caumo, Fiorella Calabrese, Federico Rea, Valentina Guarneri, Giulia Pasello

**Affiliations:** 1Division of Medical Oncology 2, Veneto Institute of Oncology IOV-IRCCS, 35128 Padua, Italy; 2Department of Radiation Oncology, Veneto Institute of Oncology IOV-IRCCS, 35128 Padua, Italy; 3Thoracic Surgery Unit, Department of Cardiac, Thoracic and Vascular Sciences and Public Health, University of Padua, 35128 Padua, Italy; 4Oncologic Radiology Unit, Veneto Institute of Oncology IOV-IRCCS, 35128 Padua, Italy; 5Department of Surgery, Oncology and Gastroenterology, University of Padua, 35128 Padua, Italy; 6Pathology Unit, Department of Cardiac, Thoracic and Vascular Sciences and Public Health, University of Padua, 35128 Padua, Italy; 7Breast Radiology Unit, Veneto Institute of Oncology-IRCCS, 35128 Padua, Italy

**Keywords:** stage III NSCLC, multidisciplinary team, multimodality treatment, neoadjuvant therapy, concurrent chemo-radiation therapy, radiotherapy, immunotherapy, PACIFIC regimen

## Abstract

**Simple Summary:**

Stage III non-small cell lung cancer (NSCLC) is a highly heterogeneous group of diseases with wide differences in tumor size and in nodal involvement and, although the intent of treatments is potentially curative, survival data still remain disappointing in some cases. The treatment of locally advanced NSCLC involves a multidisciplinary approach to determine which patients might benefit from a trimodality treatment that includes tumour resection and to identify patients with unresectable stage III NSCLC who are candidates for definitive chemo-radiation therapy (CRT). The main aim of this work was to provide a real-world description of treatment evolution and survival outcomes of stage III NSCLC patients referred to the Veneto Institute of Oncology—IRCCS and University Hospital of Padova for about 10 years.

**Abstract:**

Background: About 30% of new non-small cell lung cancer (NSCLC) cases are diagnosed at a locally advanced stage, which includes a highly heterogeneous group of patients with a wide spectrum of treatment options. The management of stage III NSCLC involves a multidisciplinary team, adequate staging, and a careful patient selection for surgery or radiation therapy integrated with systemic treatment. Methods: This is a single-center observational retrospective and prospective study including a consecutive series of stage III NSCLC patients who were referred to the Veneto Institute of Oncology and University Hospital of Padova (Italy) between 2012 and 2021. We described clinico-pathological characteristics, therapeutic pathways, and treatment responses in terms of radiological response in the entire study population and in terms of pathological response in patients who underwent surgery after induction therapy. Furthermore, we analysed survival outcomes in terms of relapse-free survival (RFS) and overall survival (OS). Results: A total of 301 patients were included. The majority of patients received surgical multimodality treatment (*n* = 223, 74.1%), while the remaining patients (*n* = 78, 25.9%) underwent definitive CRT followed or not by durvalumab as consolidation therapy. At data cut-off, 188 patients (62.5%) relapsed and the median RFS (mRFS) of the entire population was 18.2 months (95% CI: 15.83–20.57). At the time of analyses 140 patients (46.5%) were alive and the median OS (mOS) was 44.7 months (95% CI: 38.4–51.0). A statistically significant difference both in mRFS (*p* = 0.002) and in mOS (*p* < 0.001) was observed according to the therapeutic pathway in the entire population, and selecting patients treated after 2018, a significant difference in mRFS (*p* = 0.006) and mOS (*p* < 0.001) was observed according to treatment modality. Furthermore, considering only patients diagnosed with stage IIIB-C (N = 131, 43.5%), there were significant differences both in mRFS (*p* = 0.047) and in mOS (*p* = 0.022) as per the treatment algorithm. The mRFS of the unresectable population was 16.3 months (95% CI: 11.48–21.12), with a significant difference among subgroups (*p* = 0.030) in favour of patients who underwent the PACIFIC-regimen; while the mOS was 46.5 months (95% CI: 26.46–66.65), with a significant difference between two subgroups (*p* = 0.003) in favour of consolidation immunotherapy. Conclusions: Our work provides insights into the management and the survival outcomes of stage III NSCLC over about 10 years. We found that the choice of radical treatment impacts on outcome, thus suggesting the importance of appropriate staging at diagnosis, patient selection, and of the multidisciplinary approach in the decision-making process. Our results confirmed that the PACIFIC trial and the following introduction of durvalumab as consolidation treatment may be considered as a turning point for several improvements in the diagnostic-therapeutic pathway of stage III NSCLC patients.

## 1. Introduction

Lung cancer (LC) is the second-most commonly diagnosed cancer and the leading cause of cancer death in 2020, with an estimated 2.2 million new cancer cases and 1.8 million deaths, representing approximately 1 in 10 (11.4%) cancers diagnosed and 1 in 5 (18.0%) cancer-related deaths [1]. 

The lack of early onset of signs and symptoms and the heterogeneous implementation of LC screening worldwide lead to late diagnosis in most patients [2].

Non-small cell lung cancer (NSCLC) is the most common histological type (85% of all LC cases), where the majority of innovative steps forward have been taken both in systemic treatments and in radiation therapy strategies [3].

About 30% of new NSCLC cases are diagnosed at a locally advanced stage (i.e., stage III), which includes a highly heterogeneous group of patients with differences in tumour size, local extension, and pattern of nodal involvement [4,5]. The management of stage III NSCLC involves a multidisciplinary approach and a careful patient selection for surgery or radiation therapy integrated with systemic treatment [6]. Although the intent of treatments in locally advanced lung disease is potentially curative, survival data in stage III NSCLC still remain disappointing: five-year survival rates are 36%, 26%, and 13% in stages IIIA, IIIB, and IIIC, respectively [4,7]. Currently, pathological mediastinal lymph node staging is strongly recommended to establish the appropriate therapeutic strategy [8]. Surgery may be indicated for N1 or single N2 station involvement, possibly after neoadjuvant chemotherapy [9,10]; the right management of multistation N2 involvement is still a matter of debate. Treatment algorithm of stage III according to guidelines and recommendation is summarised in Figure 1.

In the 1990s, RT alone was the standard treatment for patients with unresectable NSCLC, particularly for those with N3 station involvement, although the five-year survival rate was poor (less than 10%); since then, several prospective studies have assessed the role of combined chemo-radiation therapy (CT-RT) as well as the efficacy of different CT combinations [11,12,13,14,15,16]. The RTOG 7301 was the first to establish the 60 Gy in 30 daily fractions, over five days a week for six weeks, as the standard radiotherapy dose-fractionation schedule for NSCLC [17]. In 2010, a meta-analysis on individual patient data provided by six randomised trials comparing concomitant with sequential CT-RT (cCT-RT and sCT-RT) in 1205 patients with locally advanced NSCLC showed that the concomitant combination resulted in a statistically significant 16% relative reduction in mortality, corresponding to an absolute benefit in overall survival (OS) of 4.5% and to survival rate of 15.1% at five years. However, survival benefit was associated with a significant increase in severe (grade 3–4) acute oesophageal toxicity [18]. Survival for patients with locally advanced NSCLC has improved over time: RT techniques have developed, resulting in higher conformity of the radiation fields targeting the tumour and in markedly decreased toxicities with intensity-modulated radiation therapy (IMRT) compared with 2D or 3D techniques [19]; the increased use of functional imaging such as 4D-18F-FDG PET has led to more accurate nodal staging and precise RT treatment planning [20,21].

Although the RTOG 0617 failed to demonstrate the superiority of a dose-escalated regimen (74 Gy vs 60 Gy), mainly due to higher cardiopulmonary toxicity rates [22], a secondary analysis demonstrated a comparable OS with lower incidence of side effects between the patients treated with IMRT and those with 3D techniques [23,24].

Prior to 2018, the standard of care for patients with unresectable locally advanced NSCLC was treatment with cCT-RT to a total radiation dose of 60 Gy in 30 fractions, although these patients had poor outcomes largely driven by poor distant control, as shown in Table 1 [25]. The PACIFIC trial established a new standard of care by investigating the addition of durvalumab (an anti-PDL1 monoclonal antibody) following at least two cycles of platinum-based doublet CT delivered concurrently with definitive RT to a total dose of 54–66 Gy [24]. A recent updated analysis estimated 43% of patients in the durvalumab arm remain alive at five years from randomisation and approximately one-third remain alive and free of disease progression, with an updated median OS of 47.5 vs 29.1 months (stratified HR 0.72, 95% CI 0.59–0.89) and PFS of 16.9 vs 5.6 months (stratified HR 0.55, 95% CI 0.45–0.68) [26]. In addition, consolidation therapy with durvalumab exhibited a manageable safety profile without adversely affecting patient-reported outcomes: grade 3 or 4 pneumonitis rates were comparable between the study arms. An unplanned retrospective analysis requested by the European Medicines Agency (EMA) found that durvalumab did not improve OS in a subgroup with a PDL1 expression of <1%, thus limiting the use of this drug to PDL1 positive patients [27].

## 2. Materials and Methods

This is a single-center observational retrospective (before 2018) and prospective (since 2018) study including a consecutive series of stage III NSCLC patients who were referred to the thoracic oncology multidisciplinary team of the Veneto Institute of Oncology and University Hospital of Padua (Italy) between November 2012 and November 2021, and who where considered eligible for a radical treatment. This study did not involve any change in the diagnostic and therapeutic management of patients under observation. The study design is represented in Figure 2.

The primary aim of this study was to provide a real-world description of treatment evolution of stage III NSCLC patients over ten years, with particular focus on the management shift before and after the approval of consolidation immunotherapy in Italy, in 2018. As secondary endpoints, we described treatment outcomes in terms of radiological and metabolic response in the entire study population and in terms of pathological response in patients who underwent surgery after induction therapy, relapse-free survival (RFS), and overall survival (OS). We finally explored the clinico-pathological predictors of the best outcome in each treatment subgroup.

The inclusion criteria were histologically confirmed diagnosis of stage III NSCLC, patients who were considered eligible for multimodality treatment for locally advanced disease with radical intent (CT +/− RT +/− surgery), ECOG performance status 0–1, availability of clinical and haematological data, and adequate follow-up of at least six months from the end of the radical treatment. All the cases were diagnosed through bronchoscopy with TBNA, image-guided transthoracic core needle biopsy, mediastinoscopy or open surgery. Staging was performed with brain, chest, and abdomen CT scan with iodine contrast and, in most cases, with total body 18F-FDG PET-CT scan. All patients were classified as stage IIIA, IIIB, and IIIC according to the eighth edition of the TNM staging system in Lung Cancer. The Ethics Committee of Veneto Institute of Oncology—IRCCS in Padova (CESC IOV: 2021–89) evaluated and approved the study and the informed consent which, according to the Italian Data Protection Authority dispositions, was required, whenever feasible, for collection, analysis and publication of data. Original haematoxylin-eosin (H&E) slides and matched paraffin blocks were collected before starting CT or RT and at time of radical surgery. Finally, since December 2017 the official document on the diagnostic-therapeutic pathway of NSCLC was implemented in our region, with a subsequent update in July 2022. The adherence to maps and notes of this document is currently under continuous monitoring by specific indicators (https://salute.regione.veneto.it/web/rov/polmone (accessed on 3 October 2022)).

### 2.1. Immunohistochemistry for PD-L1, ALK and ROS-1

Formalin-fixed, paraffin-embedded (FFPE) sections were processed with monoclonal antibodies anti-PD-L1 (Clone 22C3), ALK (clone D5F3), and ROS-1 (clone D4D6) using the completely automated Leica Bond III system. PD-L1 expression was evaluated by counting the percentage of positive tumor cells according to the Tumor Proportion Score (TPS). Samples were considered positive when the percentage of viable tumor cells showing partial or complete membrane staining was ≥1%. ALK and ROS-1 were evaluated using a 0–3 score system. For the assay, as a positive tissue control, normal placenta sections were used in each staining run.

### 2.2. Molecular Investigation of EGFR Mutations

EGFR mutations in exons 18–21 were tested at diagnosis if sufficient material was available. Tumor DNA was extracted from FFPE tumor slices through QIAamp DNA FFPE kit (Qiagen, Hilden, Germany); DNA sequencing was carried out with Sanger sequencing or polymerase chain reaction (PCR)-based methods (easy PGX ready EGFR kit, Diatech Pharmacogenetics, Jesi, Italy; cobas EGFR Mutation Test v2, Roche, Basel, Switzerland; EGFR mutation analysis kit EntroGen, EntroGen, Woodland Hills, CA, USA; Scorpion-ARMS EGFR Plasma RGQ PCR Kit, Qiagen, Hilden, Germany).

The percentage of residual tumour in surgical specimens was estimated by comparing the estimated cross-sectional area of the viable tumour foci with the estimated cross-sectional areas of fibrosis and necrosis (tumour bed) on each slide to evaluate the major pathological response (MPR) and pathological complete response (pCR) [28]. The widely adopted definitions of MPR and pCR assessment of resected NSCLC specimens after induction chemotherapy are less than or equal to 10% and 0% viable tumour cells, respectively [29,30,31].

Patients received systemic therapy and radiation therapy at the Veneto Institute of Oncology—IRCCS in Padova, while surgery, whenever indicated, was performed at the Thoracic Surgery Unit of the University Hospital of Padova. A multidisciplinary team, including at least a medical oncologist, a radiation oncologist, a pulmonologist and a thoracic surgeon evaluated all the patients and decided the treatment plan according to clinical practice of the reference period: radical surgery followed by adjuvant CT and/or RT; neoadjuvant CT and/or RT treatment followed by radical surgery; concurrent or sequential definitive CT-RT; patients who had disease control after CT-RT with radical intent either received consolidation durvalumab or did not depending on PD-L1 expression.

Radiological data (CT and PET-CT scan) were considered at baseline and at the end of induction treatment, that is, neoadjuvant chemo-radiotherapy or radical chemo-radiotherapy. Radiological response was evaluated according to standard RECIST criteria version 1.1 [32]. 

Where both pre- and post-treatment PET-CT scans were available, an assessment of metabolic response was carried out, determining the percentage change of maximum standardised uptake value (SUVmax). 

### 2.3. Statistical Analysis 

The overall population was evaluated for the primary aim of the study. Statistical analysis was performed using IBM Statistical Package for the Social Sciences (SPSS) software Version 26 (Inc., Chicago, IL, USA). Continuous variables were summarised by descriptive statistics, including means, standard deviations, medians, and ranges. Categorical variables were tabulated by frequency and percentage. Kaplan–Meier survival curves were calculated and *p*-values derived using a log-rank test were used to determine statistical significance of any effect observed. The Cox proportional hazards model was used to assess the impact of covariates on OS and RFS. Univariate and multivariate Cox models were analysed. The statistical significance level was set at *p* < 0.05 for all tests. OS was calculated from the date of histological diagnosis until death, patients alive at the time of analysis were censored at the date of last recorded follow-up. RFS was calculated from the date of radical surgery or beginning of CT-RT with radical intent to the date of first recurrence. Patients without recorded clinical or radiological progression were censored at the date of death (if deceased at time of analysis) or date of last recorded follow-up.

## 3. Results

The results of the study are given below: analyses on clinico-pathological characteristics, treatment pathways, and treatment outcomes performed on the entire study population.

### 3.1. Clinico-Pathological Characteristics

A total of 301 patients were included in the study: the median age of all patients was 67 years (IQR 61–72), the majority were men (*n* = 199, 66.1%) and current (*n* = 125, 41.5%) or former (*n* = 129, 42.9%) smokers. In addition, almost the entire population of interest (98%) had an Eastern Cooperative Oncology Group (ECOG) PS of less than or equal to 1. Most (*n* = 256, 85%) did not experience weight loss (weight loss greater than 10%) at diagnosis. Baseline neutrophil, lymphocyte, and platelet counts were available for most patients (*n* = 270, 89.7%): the median neutrophil-to-lymphocyte ratio (NLR) was 2.5 (IQR 1.8–3.9), and most had a low NLR (NLR < 3; *n* = 164, 54.5%), while the median platelet-to-lymphocyte ratio (PLR) was 139.1 (IQR 107.6–203.5), and low PLR (<180) was shown in most patients (*n* = 184, 67.9%). 

All patients had been diagnosed with stage III NSCLC and more than half of the patients (*n* = 170, 56.5%) had clinical stage IIIA disease according to the eighth edition of Lung Cancer TNM Classification. Most tumours were classified as T2 (*n* = 86, 28.6%), T3 (*n* = 85, 28.2%), T4 (*n* = 87, 28.9%), with N2 nodal involvement (*n*= 213, 70.8%). Most patients were diagnosed by bronchoscopy with biopsy (*n* = 152, 50.5%) and the main histotypes were adenocarcinomas (ADC, *n* = 198, 65.8%) and squamous cell carcinomas (SCC, *n* = 85, 28.2%). As regards molecular characterisation, missing data are related to the timing of introduction of the different evaluations (EGFR, ALK, and ROS1 gene alterations, PD-L1 expression). After the authorisation of admission to reimbursement of two anti-PD-1 drugs for the treatment of stage IV NSCLC, in October 2017 the Italian Association of Medical Oncology (AIOM) and the Italian Society of Pathology (SIAPEC) drafted recommendations for the immunohistochemical evaluation of PD-L1 expression as a predictive test. In locally advanced disease, PD-L1 testing on tissue biopsy at diagnosis was suggested after the introduction of consolidation immunotherapy in patients with PD-L1 ≥1% who have not progressed to CT-RT, i.e., since 2018. Among the 134 patients with known PD-L1 expression at diagnosis, most were PD-L1 positive, that is, ≥1% (*n* = 74, 55.2% of tested population) and in the overall population 24 patients (8%) were oncogene-addicted because of a driver alteration in *EGFR*, *ALK*, or *ROS1* genes. The patients’ characteristics and clinical findings are summarised in Table 2.

### 3.2. Therapeutic Pathways and Treatment Responses

Patients received a multimodal therapy with radical intent, through a combination of strategies among surgery, chemotherapy, radiotherapy, and consolidation immunotherapy. The majority of patients received surgical multimodality treatment (*n* = 223, 74.1%); in particular, 119 (39.5%) patients underwent neoadjuvant therapy followed by surgery, while the remaining patients (*n* = 78, 25.9%) underwent definitive CT-RT followed or not by durvalumab as consolidation therapy. Radiation therapy was administered concurrently with CT in more than half of the cases (*n* = 45, 57.7%). Among patients who underwent upfront surgery (*n* = 104), the majority had clinical stage IIIA disease (*n* = 82, 78.8%) and 22 (21.2%) patients had N1 nodal involvement while 66 (63.5%) N2. One-hundred and nineteen patients received induction therapy before surgery and the majority had clinical stage IIIA disease (*n* = 67, 56.3%), and fourteen (11.8%) patients had N1 nodal involvement while ninety-nine (83.2%) N2. Finally, among patients who underwent CT-RT (*n* = 78), the majority had clinical stage IIIB-C (*n* = 57, 73.1%), 48 (61.5%) patients had N2 nodal involvement, and 26 (33.3%) N3. All treatment modalities and treatment responses are summarised in Table 3. The discrepancy between the number of patients who underwent neoadjuvant therapy (*n* = 122) and those who actually underwent surgery after induction treatment (*n* = 119) is due to the exclusion from the radical surgical approach of three candidates due to progression to neoadjuvant therapy resulting in a change in the final treatment pathway, as can be seen in the consort flow diagram (Figure 1).

The year of the greatest changes in the treatment of patients with unresectable stage III NSCLC who were candidates for CT-RT was 2018, due to the improvements in RT planning techniques, dose constraints, and treatment delivery that allowed more patients to receive concomitant treatment and adequate doses, and because of the introduction of consolidation immunotherapy into clinical practice. As a consequence of a more correct staging at diagnosis and an increase over time in patients who were candidates for radical CT-RT, the percentage of patients who were candidates for neoadjuvant treatment in order to achieve resectability decreased: before 2018 we observed a percentage of neoadjuvant treatment followed by surgery of 47.1% among all treatment modalities, whereas after 2018 the percentage dropped to 32.9%, while the percentage of patients undergoing CT-RT increased from 14.3% to 36.1% and this growth is driven by the proportion of patients who received CT-RT followed by durvalumab (19.3%) (see Figure 2). Furthermore, before 2018 only 30% of patients (*n* = 6) received concomitant CT-RT, whereas afterwards 67.2% received concomitant treatments (*n* = 39) (see Figure 3).

In the intention to treat (ITT) population, 122 patients underwent neoadjuvant therapy and the radiologic evaluation at the end of the induction treatment revealed the following data: 65 patients (53.3%) had a partial response (PR), 1 (0.8%) had a complete radiological response (CR), 51 (41.8%) reached a stable disease (SD), and 4 (3.3%) had progressive disease (PD). 

As regards the pathological response, the evaluation of tumour bed in surgical specimens of 70 patients who underwent neoadjuvant therapy has been made to assess the MPR and pCR. In our study population, 46 ADC, 19 SCC, 3 sarcomatoid carcinomas, 1 adeno-squamous carcinoma, and 1 not-otherwise-specified carcinoma were included. The median rate of viable tumour was 55% (IQR 20–70): 12 (17.1%) patients reached MPR, and among them 6 (8.6%) were pCR. According to the proposed thresholds across different histologies by Qu et al., we observed 4 (21.1%) MPR and 2 (10.5%) pCR in patients with SCC (N = 19); 34 (66.7%) MPR and 4 (7.8%) pCR were observed among non-SCC patients (N = 51) [33].

Among the 78 patients who underwent CT-RT with radical intent, 45 (57.7%) achieved a PR, 17 (21.8%) had SD, and 14 (17.9%) experienced PD.

### 3.3. Treatment Outcomes

The median follow-up of the ITT population was 27.8 months (IQR 15.5–50.2). At data cut-off (31 May 2022), among the overall population of 301 patients, 188 patients (62.5%) relapsed and the median RFS (mRFS) of the entire population was 18.2 months (95% CI: 15.83–20.57) (see Figure 4).

At the time of analyses 140 patients (46.5%) were alive, while 161 patients (53.5%) were dead and the median OS (mOS) was 44.7 months (95% CI:38.4–51.0) (see Figure 5). In our ITT population, 2- and 5-year survival rates were 68% and 32%, respectively. In IIIA patients, the 2-year survival rate was 71% while in IIIB-C the 2-year survival rate was 64%. 

We analysed the outcomes as per radical treatment modality: both the mRFS and the mOS were significantly different among subgroups (*p* = 0.002 for RFS analysis and *p* < 0.001 for OS analysis) according to the therapeutic pathway. Patients receiving surgery followed by adjuvant treatment and CT-RT followed by durvalumab had better outcomes (see Figure 6 and Figure 7). Outcomes as per treatment modality are described in Table 4.

In the overall population, there was no significant difference in terms of mRFS (*p* = 0.694) between patients treated before and after 2018 in the overall population: mRFS of 18.2 months (95% CI: 14.43–21.97) and 18.2 months (95% CI: 13.13–23.27), respectively, before and after 2018. Additionally, for mOS there was no significant difference (*p* = 0.887) between the two periods: mOS of 42.5 months (95% CI: 33.07–52.02) and 46.5 months (95% CI: 37.59–55.51), respectively, before and after 2018.

Outcome differences according to the treatment pathway were confirmed selecting patients treated after 2018 but not before 2018; indeed, a significant difference in mRFS (*p* = 0.006) and mOS (*p* < 0.001) was observed according to treatment modality, as shown in Figure 8 and Figure 9. Table 5 reports outcomes as per initial treatment strategy in patients treated after 2018.

Stratifying patients by stage at diagnosis of disease, considering only stage IIIA patients (*n* = 170, 56.5%), no difference in mRFS (*p* = 0.054), but a significant difference in mOS was evidenced (*p* = 0.038) as per the treatment algorithm; in particular, patients who received surgical multimodality treatment had better outcomes than those undergoing radical CT-RT. Moreover, selecting stage IIIA patients treated after 2018 (*n* = 86, 28.6%), a more remarkable statistical significance was found in terms of mOS (*p* < 0.001) according to treatment strategies (see Figure 10 and Figure 11).

Furthermore, considering only patients diagnosed with stage IIIB-C (*n* = 131, 43.5%), there was a significant difference both in mRFS (*p* = 0.047) and in mOS (*p* = 0.022) as per the treatment algorithm (see Figure 12 and Figure 13).

Additionally, selecting stage IIIB-C patients treated after 2018 (*n* = 75, 24.9%), a statistical significance was found in terms of mRFS (*p* = 0.047) and mOS (*p* = 0.024) according to treatment strategies (see Figure 14 and Figure 15). In detail, in terms of mRFS, better outcomes were achieved by surgery followed by adjuvant treatment and CT-RT followed by consolidation durvalumab. In terms of mOS, despite the immaturity of the data, the best outcome was achieved by the CT-RT treatment plus durvalumab arm.

Considering the subset of unresectable stage III NSCLC patients (*n* = 78) who underwent CT-RT as therapy with radical intent, we analysed outcomes by distinguishing patients who underwent only CT-RT (*n* = 47, 60.3%) from those treated with CT-RT followed by consolidation durvalumab (*n* = 31, 39.7%). The mRFS of the whole unresectable population was 16.3 months (95% CI: 11.48–21.12), with a significant difference among subgroups (*p* = 0.030) in favour of patients who underwent the PACIFIC regimen. The mOS of the whole unresectable population was 46.5 months (95% CI: 26.46–66.65), with a significant difference between two subgroups (*p* = 0.003) in favour of consolidation immunotherapy. See Figure 16 and Figure 17.

Outcomes according to the treatment modality are described in Table 6. 

Furthermore, considering patients treated with CT-RT after the introduction of immunotherapy as consolidation therapy (*n* = 58), a more statistically significant difference was observed in terms of mRFS (*p* = 0.002) and mOS (*p* < 0.001) between two subgroups, as shown in Figure 18 and Figure 19.

## 4. Discussion

The treatment algorithm for resectable and unresectable stage III NSCLC has been rapidly changing, thus impacting the multidisciplinary team decision-making process. A wide overview of the improvement in the diagnostic-therapeutic pathway of this heterogeneous group of patients in recent years may be useful to establish a prompt response to the upcoming innovative strategies.

The optimal management of stage III NSCLC is the result of two key-points: accurate clinical and pathological staging and the subsequent definition of the therapeutic strategy upfront by the multidisciplinary team discussion. In particular, the pathological staging of mediastinal lymph nodes is recommended, in order to better select patients eligible for surgery or CT-RT approach [8].

Despite the limitations related to the retrospective observational nature of data collection, our work provides insights into the management and the survival outcomes of stage III NSCLC patients in the real world, with a median follow-up of 27.8 months on a wide single-center population treated over ten years.

We reported the 2- and 5-year survival rates of 68% and 32%, respectively, in the overall population; the 2- and 5-year survival rates were 71% and 34%, respectively, in stage IIIA, while 64% and 30%, respectively, in stage IIIB-C. Estimates of survival probability in our study population met literature data [4,7], thus reinforcing the quality of data collection and analysis, although retrospective. This was also clearly suggested when we focused on outcomes of patients undergoing CT-RT (*n* = 47, 60.3%) or of patients treated with CT-RT followed by consolidation durvalumab (*n* = 31, 39.7%). In the PACIFIC trial the estimated 2-year OS was 66.3% for the durvalumab arm and 55.3% for the placebo, respectively [26]. In our case series, a significant difference between subgroups was evidenced both in terms of mRFS (*p* = 0.030) and mOS (*p* = 0.003) in favour of consolidation immunotherapy; indeed, we estimated a 2-year OS rate of 75% in patients receiving durvalumab and 39% in those who did not receive any systemic consolidation treatment, respectively.

Our results confirmed that the PACIFIC trial and the following introduction of durvalumab as consolidation treatment may be considered as a turning point for several improvements in the diagnostic-therapeutic pathway of stage III NSCLC patients: the increasing rate of cCT-RT over the sequential approach and the reduction of the subgroup of patients receiving neoadjuvant chemotherapy followed by surgery are direct results of the improved survival offered in the consolidation setting, as well as the improvement of nodal staging. Indeed, in our series, outcome differences according to treatment pathways were confirmed by selecting patients treated after 2018 but not before 2018, thus underlining the contribute of cCT-RT followed by durvalumab, and of a more accurate selection of patients eligible for surgery in the latest years. Furthermore, the innovative RT features, such as IMRT and a rigid IGRT protocol, lead to better plan conformity and dose delivery, thus improving the capability to perform concomitant chemo-radiotherapy treatments. Finally, elective nodal irradiation of the mediastinum is no more the standard in concomitant RT planning and this allows sparing nodal stations not involved. This could be useful when radical chemoradiation is followed by consolidation immunotherapy.

Additionally, in the observational PACIFIC-R study, in which the real-world effectiveness of durvalumab was assessed in patients from an early access program, as expected, real-world PFS was higher among patients who received cCT-RT versus sCT-RT and patients with higher PD-L1 expression [34] 

Recently, at the World Conference on Lung Cancer 2022, Stirling et al. presented data from the Australian Lung Cancer Registry on 1396 stage III NSCLC patients consecutively treated from 2012 to 2019. In this real-world study, 67% of patients received radical treatment with a mOS of 38.0 months. Surgical approach decreased over time, while the use of cCT-RT and immunotherapy increased. The authors underlined that durvalumab improved the survival of CT-RT, producing similar survival outcomes to that of surgical combinations within first three years after diagnosis [35]. Other series reported a worse outcome for those patients considered eligible for neoadjuvant treatment and eventually converted to CT-RT, compared with those immediately addressed to a CT-RT strategy [36].

In a Dutch multicenter analysis retrospectively conducted on 855 locally-advanced NSCLC patients, the authors underlined that between 2015–2017 and 2018–2019, the proportion of patients undergoing cCT-RT increased from 34% to 42% (*p* = 0.02) and the use of sCT-RT declined (21% to 16%, *p* = 0.05) [37]. A different distribution of stage III NSCLC patients into different treatment groups emerged also from our data, where we observed a 14.2% reduction of patients undergoing neoadjuvant chemotherapy followed by surgery after 2018, with an increase of 21.8% of the overall subgroup receiving CTRT and of 37.2% of those planned for concomitant despite sequential approach. It is tempting to speculate that this strategy shift could be also the result of the decreasing rate of postoperative mediastinal radiation therapy, in light of the results of the recent Lung ART and PORT-C trials [38,39].

Patients’ outcomes after induction chemotherapy and surgery are directly related to the MPR and pCR rates; however, available data showed a pCR only in about 5% of the cases after platinum-based chemotherapy, while the percentage of cases achieving a MPR was more heterogeneous [31]. The widely adopted definition of MPR assessment of resected NSCLC specimens after induction chemotherapy is less than or equal to 10% viable tumour. However, in a study by Qu et al., the authors demonstrated that pathologic response following neoadjuvant therapy and subsequent survival outcomes differ between histologic types of NSCLC: MPR (≤10% viable tumour) predicted better survival in patients with SCC, while in patients with ADC, the optimal threshold was 65% [33]. In our case series, we observed a pCR and a MPR in 8.6% and 17.1% of patients, respectively. Some discrepancies in terms of pathological responses may be explained by the heterogenous histologic subgroups represented in different series; indeed, a different threshold of residual viable tumour has been proposed in responders with squamous and non-squamous histologies [40]. Moreover, different types of neoadjuvant therapies induce heterogeneous effects in the tumour microenvironment, thus leading to different thresholds for defining the MPR and to a more complex pathological evaluation and reporting. Indeed, the concept of the tumour bed as a whole and the pathological report of the regression bed and necrosis together with residual tumour cells have been introduced in patients receiving induction immune-checkpoint inhibitors [28,41]. The increasing percentage of pCR (15% to 38%) and MPR (18% to 57%) and the increasing rate of patients receiving a more conservative surgery after neoadjuvant chemotherapy combined with immunotherapy [42,43,44] should be considered in the near future, when the proportion of stage III NSCLC patients eligible for radical surgery may be reconsidered.

In this evolving landscape, some medical needs are still unmet, such as the identification of prognostic and predictive biomarkers besides PD-L1 expression; inflammatory signatures in blood and tumour samples, tumour immune microenvironment characterisation, and radiomic features should be integrated with clinico-pathological features of NSCLC patients [45,46] 

## 5. Conclusions

The optimal management of stage III NSCLC patients should be based on a multidisciplinary team discussion and adequate staging in order to depict upfront the optimal treatment pathway for each patient. Innovation coming into radiation therapy and systemic treatment, such the introduction of consolidation immune-checkpoint inhibitors, has impacted the diagnostic-therapeutic management and on a treatment shift over the years, particularly for those patients in stage IIIB-C. These real-world data show a dynamic scenario in the management of locally advanced NSCLC, which may be further improved in the future by the upcoming innovation in neoadjuvant and adjuvant systemic treatment. The promptness of the multidisciplinary team to optimise the different steps of the patients’ journey will hopefully translate into the improvement of patients’ prognosis and quality of life.

## Data Availability

The data presented in this study are available on request from the corresponding author.

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
