# Peer review of "The Multidisciplinary Approach in Stage III Non-Small Cell Lung Cancer over Ten Years: From Radiation Therapy Optimisation to Innovative Systemic Treatments"

_cancers, 2022, doi:10.3390/cancers14225700_

Round 1
Reviewer 1 Report
In this single-center observational retrospective and prospective study, the authors analyzed consecutive 301 stage III non-small cell lung cancer patients regarding their demographic data, cancer characteristics, treatment, major pathological response (MPR) and pathological complete response (pCR), relapse-free survival (RFS), and overall survival (OS). In this study, major difference and shift of treatment paradigm were found after introducing Durvalumab for consolidation in 2018. The authors also advocated using different strategies for stage III subgroups, namely IIIA, IIIB, and IIIC.
Minor questions:
1. Patients referred to the Veneto Institute of Oncology IRCCS and University Hospital of Padova for multimodality treatment were included. Therefore, we did not find any patients received only best supportive care, chemotherapy only, or radiotherapy only. Besides, the ECOG statuses at diagnosis were mostly 0 (44.2%) and 1 (53.8%). Hence, this study is not true real-world description of treatment for all stage III NSCLC patients.
2. For Table 3, revision for listing is recommended. Besides, 89 patients received adjuvant therapy including 85 for adjuvant systemic therapy. But, 101 received adjuvant radiotherapy was mentioned?
3. Figures 18 & 19 described those who were treated after 2018. In both Figures 18 & 19, the survival curves for radical CT-RT group extended to 60 months (5 years)?
Author Response
point-by-point response to the reviewer:
- We thank the reviewer for this observation, we integrated the methods section by adding the eligibility criterium that all patients were considered eligible for treatment with a radical intent, ECOG PS 0-1. Although with those eligibility criteria, the study is a real world report about the multidisciplinary treatment approach in those patients. Moreover, in the consort flow diagram, we underlined that 46 (13.3%) of 347 stage III NSCLC were not included in our work. Among them patients not suitable for treatment with radical intent (for PS, comorbidities, cardiopulmonary function…) were candidate only to palliative treatment or best supportive care.
-
We thank the reviewer for this observation, we revised Table 3 accordingly. Some patients who underwent the induction systemic therapy before surgery also received adjuvant radiotherapy. Therefore, the number of adjuvant radiotherapies is 101 because 55 (54.5%) patients after neoadjuvant treatment and surgery also underwent adjuvant radiotherapy (the other 56 patients are within the group of 119 patients).
-
We thank the reviewer for this question. Overall survival was calculated from the date of histological diagnosis until death, patients alive at the time of analysis were censored at the date of last recorded follow-up. At data cut-of (May 31, 2022), one patient who started CRT in 2018 but was diagnosed with NSCLC in 2017, had an OS of 60 months.
Reviewer 2 Report
The treatment paradigm for stage III NSCLC patients before and after the introduction of durvalumab was drastically changed, and this trend is actually reflected in the realworld. It is considered that this study showed change of treatment paradigm in real world and actually led to better clinical outcomes in stage III NSCLC patients.
1. In my clinical experience as well as previous reports (Cancer Res Treat. 2019;51(2):493-501), it is known that the PFS is shorter in patients with stage III NSCLC treated CCRT compared to those with EGFR wild or unknown. What is the proportion of EGFR mutation positive patients in your study, and how does the PFS of EGFR mutation positive patients after surgery or CCRT compared to those without?
2. In selecting patients eligible for CCRT or surgery, pulmonary function is also important because the underlying lung function plays an important role to predict treatment complications such as radiation pneumonitis and respiratory deterioration after CCRT or surgery. In your study, the proportion of former and current smokers were considerable. what is the proportion of patients with chronic respiratory disease such as COPD or IPF?
Author Response
point-by-point response to the reviewer:
-
We thank the reviewer for this comment; however as described in table 1, our case series reported only a 8% of oncogene addicted NSCLC patients (EGFR, ALK or ROS1 positive), thus a comparison between the two subgroups would not be well balanced and any conclusion about treatment effectiveness in EGFR+ and EGFR WT or unknown would not be possible from our case series. Moreover, we think that, as any guidelines underline, when there is a place for treatment with radical intent in locally advanced stage, the multidisciplinary team should not exclude cCRT in oncogene addicted tumors.
-
We thank the reviewer for this observation. We verified that 112 patients in our study population had a positive history of COPD. We added this data to the table about clinicopathological characteristics (Table 2).
Reviewer 3 Report
The manuscript is well written and investigates a clinically meaningful topic. Moreover, it presents a cohort of patients with locally advanced NSCLC in which about 40% of patients underwent neoadjuvant chemotherapy followed by surgery. This is particularly interesting due to the lack of data about neoadjuvant chemotherapy in real-life setting.
However there does not seem to be any information on the changes that have taken place in the Center regarding the integrated care pathways.
In particular, it would be interesting to know if the treatment pathways have been formally adopted, if some process indicators have been measured and what evolution has taken place over time.
Overall, I believe that the manuscript needs some revisions before being suitable for publication.
1-First of all, I found the title not perfectly coupled with the manuscript since the text did not describe how the management has changed in the past years as the title would suggest. You described this change just briefly in the discussion.
2-The authors did an amazing work summarizing all the evidence regarding locally advanced NSCLC but I believe that for the sake of the paper introduction should be shortened.
3-Why do you state that in 2018 also the RT treatment paradigm and constraints changes?Please provide a reference.
4-Could you please provide more details of the subgroup of patients you presented: e.g nodal status of patients undergone neoadjuvant chemotherapy?
5-Survival analyses: the authors compared the survival of patients according to treatment modality (figure 6 and followings). I am not sure this is a correct approach since the treatment modality is mainly determined by the stage of the disease which obviously itself influences survival. Same goes for the stage IIIA subgroup: different nodal stages may explain different prognosis. please adjust survival analyses.
6-it seems that some patients with stage IIIB-C underwent surgery (line 390 and following): please explain that.
Some more specific comments below
-line 87: the reference you cite (Postmus et al) clearly states that surgery is recommended -outside of clinical trial- only for single station N2. Please revise accordingly
-scheme 1: please provide references (e.g. ESMO, NCCN guidelines) and specify drug approval setting (e.g. Durva is reimbursed also for PDL-1 neg in extra EU countries)
-line 115: treatment planning and nodal staging are two different steps. Please rephrase.
- Results: I suggest to move to the discussion section the part regarding treatment changes after 2018, the part regarding missing molecular data, the Qu et al comparison
I would suggest the authors to add the following meaningful references:
- 10.1016/j.annonc.2022.02.003
- 10.1016/j.annonc.2022.06.013
- 10.1016/j.ejca.2022.02.015.
- 10.1016/j.lungcan.2021.03.013
- PACIFIC R update, JTO 2022
Author Response
We thank the reviewer for rising this issue, which is really important for our management of NSCLC and other solid tumors in the Veneto Region. According to this comment, we integrated the method section by mentioning our regional document on the diagnostic-therapeutic pathway of NSCLC. The monitoring of the specific indicators is currently ongoing and it is not possible to publish the results of this monitoring over time. However, the reader may find the document and indicators/benchmarks on the website indicated (https://salute.regione.veneto.it/web/rov/polmone).
Point-by-point response to the reviewer:
-
Thanks to the reviewer for this suggestion, we changed the title accordingly into “The multidisciplinary approach in Stage III Non-Small Cell Lung Cancer over ten years: from Radiation Therapy optimization to innovative Systemic Treatments”
-
We agree with the reviewers suggestion, and we shortened the introduction section accordingly
-
Thank you for this comment; in the introduction section we mentioned both the PACIFIC trial and the RTOG 0617 trial which were published at the end of 2017 (references 23,24); thus it is reasonable to state that since 2018 the Italian radiation oncology teams transposed ta new paradigm in terms of timing of chemo and radiotherapy and in terms of IMRT implementation. These issues coincide with the experience of our Center where we observed a technical and personnel improvement between 2017 and 2018.
-
Thanks for your observation. We added some specific data in the main text, in the section “3.2 Therapeutic pathways”.
-
We thank the reviewer for this comment. In the section about “Therapeutic pathways” we specified clinical stage disease and nodal involvement according to the therapeutic pathway to underline differences among subgroup of patients. Concerning survival analyses, we first consider all patients and estimated RFS and OS according to treatment modalities and then, we consider only patients diagnosed with stage IIIA or IIIB-C.
Moreover, in order to clarify the impact of the T and N parameters on patients outcome, we performed a multivariate analysis (data not shown) using the stage as covariate and the only features which impacted on overall survival were the presence of oncogene addition; the use of immunotherapy in the consolidation setting; and finally a partial or complete response to chemo-radiotherapy.
-
We added some specific data in the main text, in the section “3.2 Therapeutic pathways”.
As regard the specific comments:
- line 87: Thanks to reviewer, we modified the text accordingly.
- scheme I: Thanks for your observation. We modified Scheme I according to your comment.
- line 115: Thanks to reviewer, we modified the text accordingly.
As regard the discussion, we thank for your observation. We modified the discussion according to your suggestion.
Moreover, we have add some meaningful references.
Round 2
Reviewer 2 Report
The reviewer's request was answered appropriately in revised manuscript.